# Comparison among Activities and Isoflavonoids from *Pueraria thunbergiana* Aerial Parts and Root

**DOI:** 10.3390/molecules24050912

**Published:** 2019-03-05

**Authors:** Eunjung Son, Jong-Moon Yoon, Bong-Jeun An, Yun Mi Lee, Jimin Cha, Gyeong-Yup Chi, Dong-Seon Kim

**Affiliations:** 1Herbal Medicine Research Division, Korea Institute of Oriental Medicine, 1672 Yuseong-daero, Yuseong-gu, Daejeon 34054, Korea; ejson@kiom.re.kr (E.S.); candykong@kiom.re.kr (Y.M.L.); 2Division of Bio-Technology and Convergence, Daegu Haany University, 285-10 Eobongji-gil, Gyeongsan, Gyeongsangbuk-do 38578, Korea; yjm2777@naver.com (J.-M.Y.); anbj@dhu.ac.kr (B.-J.A.); 3Department of Microbiology, Faculty of Natural Science, Dankook University, Cheonan, Chungnam 31116, Korea; jimincha@dankook.ac.kr

**Keywords:** *Pueraria thunbergiana*, phenolic content, NO, iNOS, COX-2, kudzu leaves

## Abstract

Kudzu (*Pueraria thunbergiana* Benth.) has long been used as a food and medicine for many centuries. The root is the most commonly used portion of the plant, but the aerial parts are occasionally used as well. In this study, we investigated the constituent compounds and biological activities of the aerial parts, leaves, stems, and sprouts, and compared their constituents and activities with those of roots. Leaf extract showed a significantly higher TPC level at 59 ± 1.6 mg/g and lower free radical scavenging (FRS) values under 2,2-diphenyl-1-picrylhydrazyl (DPPH), 2,2’-azino-bis(3-ethylbenzothiazoline-6-sulphonic acid (ABTS), and NO inhibition at 437 ± 11, 121 ± 6.6 μg/mL and 107 ± 4.9 μg/mL, respectively, than those of sprout, stem, and root extract. Leaf extract also significantly suppressed lipopolysaccharide (LPS)-mediated gene expression of inducible nitric oxide synthase (iNOS) and cyclooxygenase-2 (COX-2). The main components of leaf extract were found to be genistin and daidzin. This study suggests that the leaves of kudzu are a good source of biological activities and isoflavones that can be used in functional or medicinal foods and cosmetics for the prevention or treatment of diseases related to inflammation and oxidative stress.

## 1. Introduction

Kudzu (*Pueraria thunbergiana* Benth.) is native to Korea, Japan, China, and India [1] and is now grown worldwide. Leaves, buds, and sprouts of kudzu are listed as foods in Korea and are available for foods that have been consumed as a tea, salad, juice, jelly, pickles, and syrup in many countries. Kudzu roots are used as a source of starch in Japan and China and are eaten as a vegetable [2]. Kudzu roots have been commonly used as a food and medicine for centuries, especially in Asia [3], and reported to control diabetes, prevent cardiovascular diseases [4], and have antioxidant, anti-hypertensive [5], and anti-inflammatory activities [6]. The active constituents of the plant mainly include isoflavonoids, triterpenoid saponins, chalcones, and coumarins [3]. Puerarin (4’,7’-dihydroxy-8-β-D-glucosylisoflavone), daidzin, and daidzein are known to be the main flavonoids present in kudzu [7,8]. Puerarin, which is a major component of kudzu roots, has a variety of biological activities, including activity against osteoporosis [9], cardiovascular disease [10], gynecological diseases [11], cognitive impairment [12], and diabetic nephropathy [13], and it can serve as an antioxidant. Daidzin and daidzein were reported to have osteogenic [14], antiestrogenic [15], and cholesterol-lowering [16] activities.

Kudzu roots are used commercially in various products, including teas, soaps, dietary supplements, gums, and cosmetics, but its aerial parts are rarely used and studied. Therefore, we investigated the constituents and biological activities of the aerial parts, namely leaf extract (KL), stem extract (KST), and sprout extract (KSP), and compared them with those of the root extract (KR) to evaluate their versatility and applicability.

## 2. Results and Discussion

### 2.1. Total Phenolic Contents

In the present study, KL, KST, KR, and KSP were evaluated using the Folin–Ciocalteu method [17]. The contents varied from 15 mg/g to 63 mg/g with KR (63 ± 3.7 mg/g) and KL (58 ± 1.6 mg/g) having significantly higher total phenolic compound (TPC) levels than KSP (38 ± 0.8 mg/g) and KST (25 ± 3.0 mg/g).

Phenolic compounds such as flavonoids and tannins were considered to be the major contributors to the antioxidant activity of these plants. These compounds are ideal antioxidants because they are highly reactive as both electron and hydrogen donors, and they are also capable of chelating metal ions [18]. These antioxidants also possess diverse biological activities, such as anti-inflammatory, anticarcinogenic, antiatherosclerotic, and antiaging effects. Phenolic compounds are ubiquitous in plants used for foods, medicines and cosmetics; they occur as glycosides and have several phenolic hydroxyl groups. They are known for their efficient radical scavenging activity resulting from the hydroxyl groups at various positions and the ortho-dihydroxy structure in their B ring [19]. Recent investigations have shown that phenolic compounds contribute significantly to the antioxidant activities of many fruits, vegetables, and medicinal plants [20]. Kudzu roots have been shown to contain large amounts of isoflavones (an average of 1.8 to 12% dry matter), including puerarin, daidzin, daidzein, rutin, caffeic acid, gallic acid, chlorogenic acid, quercetin, quercitrin, hyperosidem, rhamnetin, kaempferol, myricetin, *p*-coumaric acid, ferulic acid, sinapic acid, and *p*-hydroxybenzoic acid [21]. From this study, KR and KL showed total phenolic contents of 6.3% and 5.8%, respectively. This result suggests that both Kudzu leaf and root extracts are rich sources of phenolic compounds.

### 2.2. Antioxidant Activity

DPPH free radical scavenging (FRS) activity assays are widely used for screening medicinal plants to investigate their antioxidant potential. [22]. The DPPH FRS activities were evaluated, and the results are shown in Table 1. KL showed potent antioxidant activity, and the FRS_50_ values of the extracts increased in the following order: KL (436 ± 10.9 μg/mL) > KR (582 ± 16.4 μg /mL) > KSP (755 ± 8.6 μg/mL) > KST (1,136 ± 14.2 μg/mL).

The ABTS^+^• scavenging activity assay is a simple method for evaluating the activity of the ABTS^+^• hydrogen atom or electron abstraction from the compounds under study [23]. ABTS^+^•, a cationic free radical soluble in both water and organic media, is produced by reacting ABTS with potassium persulfate [24]. The differences in the ABTS^+^• scavenging activities of each sample, the control, and EGCG are presented in Table 1. The FRS_50_ value for each compound for ABTS^+^• increased in the following order: KL (121 ± 6.6 μg/mL) > KR (138 ± 2.7 μg/mL) > KSP (342 ± 1.4 μg mL) > KST (455 ± 17.1 μg/mL).

The xanthine oxidase inhibition activities were evaluated at 200 μg/mL, and the inhibition activities decreased in the following order: KR (52 ± 2.8%) > KL (37 ± 1.6%) > KST (4.6 ± 0.7%) > KSP (4.0 ± 2.1%).

In summary, KL and KR showed similar antioxidant activities while the values of KSP and KST were lower, and these activities were well matched with the total phenolic contents determined in this study. The results suggested that like the roots, kudzu leaves could also be used as a source of antioxidant and antiaging compounds.

### 2.3. Cell Viability

To ensure that the cells were healthy before being subjected to the bioactivity assays, and that the extract at the relevant concentrations was not toxic to the cells, the cell viability was measured after treatment with various concentrations of the kudzu extracts from different parts of the plant. The viability of RAW 264.7 cells decreased upon exposure to 200 µg/mL of KL, 400 µg/mL of KSP, 1000 µg/mL of KST and KR as shown in Figure 1. Thus, the concentrations of the kudzu extracts selected for subsequent studies were in the range of 5 to 100 µg/mL for KL, 5 to 200 µg/mL for KSP, and 100 to 500 µg/mL for KST and KR.

### 2.4. Effect of Kudzu Extracts by Part on NO Production

The inducible forms of NOS are the proinflammatory enzymes most responsible for increasing the levels of NO. Therefore, the effect of tested samples on the inhibition of NO production and the inducible nitric oxide synthase (iNOS) level were investigated. NO in the biological matrix is very unstable and rapidly oxidizes to nitrite (NO^2−^), and thus the measurement of nitrite is routinely used as an indicator of NO production.

In the untreated RAW 264.7 cells, the concentration of nitrite could not be detected (Figure 1). Once the cells were stimulated with LPS, a large amount of nitrite was produced. The IC_50_ values for NO production increased in the following order: KL (107 ± 4.9 μg/mL) > KST (202 ± 7.9 μg/mL) > KSP (293 ± 11.1 μg/mL) > KR (354 ± 0.7 μg/mL). The leaf extract of kudzu showed the strongest inhibition activity toward nitrite production.

### 2.5. Anti-Inflammatory Activity of Kudzu Extracts by Part

The anti-inflammation activities of the extracts prepared from leaves, stems, sprouts, and roots were investigated. Inhibition of iNOS and COX-2 protein expression by KL was evaluated for extract concentrations ranging from 10–100 μg/mL. Treatment with KL above 10 μg/mL significantly inhibited iNOS and COX-2 expression (Figure 2). KR was evaluated at concentrations from 50–500 μg/mL and treatment above 50 μg/mL significantly inhibited iNOS and COX-2 expression (Figure 3). Each extract inhibited iNOS and COX-2 expression in a dose-dependent manner.

Macrophages are generally an important component in the immune defense mechanism. During inflammation, macrophages actively participate in inflammatory responses by releasing proinflammatory cytokines and mediators [25]. Furthermore, proinflammatory mediators such as NO, iNOS, and COX-2 play key roles in the pathogenesis of many acute and chronic inflammatory conditions [26]. The iNOS and COX-2 pathways are known to play an important role in inducing ROS production [27]. First, we determined the effects by part of kudzu. Pretreatment of cells with KL, KST, KSP, and KR significantly reduced NO production. Overproduction of NO due to overexpression of iNOS has been implicated in the pathogenesis of inflammation, carcinogenesis, and septic shock [28]. COX-2 is another inducible enzyme that catalyzes the biosynthesis of PGE_2_, which contributes to the pathogenesis of various inflammatory diseases, including invasion, edema, angiogenesis, and tumor growth [29]. Thus, the anti-inflammatory agents that decrease NO production by simultaneously inhibiting iNOS and COX-2 expression may have potential therapeutic effects for the treatment of inflammatory and infectious diseases. According to our results, kudzu extracts strongly inhibit LPS-induced NO production by attenuating the protein expression of iNOS and COX-2 without notable cytotoxicity. Several plant-derived components, including resveratrol, curcumin, isoflavones, and red ginseng oil, have been reported to inhibit iNOS and COX-2 expression, but the anti-inflammatory effects of kudzu extracts, especially from the aerial parts, have not been reported. The results of this study clearly showed that the kudzu extracts can inhibit NO production through the inhibition of both iNOS and COX-2 expression. Among the extracts, kudzu leaf extract showed more potent effects than root extracts, which is the only part of the plant currently being used in the production of food and cosmetics. This result means the leaves of kudzu may be a better source of potent anti-inflammatory compounds than roots.

### 2.6. HPLC for Quantitative Analysis

An analysis method for quantifying seven compounds from kudzu extracts by parts was developed. The compounds were identified by a comparison of their experimental retention times, UV spectra and mass spectrums to those of reference standards (Figure 4). Each 7 components were confirmed on the positive ionized mass spectrum as follows; puerarin ([M+H]^+^ 417.29), daidzin ([M+H]^+^ 417.32), schaftoside ([M+H]^+^ 565.41), genistin ([M+H]^+^ 433.29), ononin ([M+H]^+^ 431.31), daidzein ([M+H]^+^ 255.17), genistein ([M+H]^+^ 271.14). According to the profile of the corresponding UV spectra, 245 nm was determined to be most appropriate detection wavelength for the quantification analysis on these compounds. The concentration of the 7 compounds are shown in Table 2 and Figure 5. KR contained 67 ± 2.2 mg/g of puerarin, 4.3 ± 0.3 mg/g of genistin, and 2.0 ± 0.2 mg/g of ononin, which is consistent with previous reports [30]. KL showed high quantities of isoflavones but mainly genistin (61 ± 1.1 mg/g) and daidzin (5.5 ± 0.3 mg/g). The total isoflavone content decreased in the following order: KR (73 ± 2.7 mg/g) > KL (66 ± 1.4 mg/g) > KST (2.3 ± 0.2 mg/g) > KSP. As already reported, the major isoflavone of kudzu root extract was puerarin, but the major isoflavone of kudzu leaf extract was genistin. This is the first report of genistin being detected in kudzu. Genistin has been reported to have biological activities including mucopolysaccharidose reducing [31], estrogenic potency [32], and hypocholesterolemia lowering activities [32]. Kudzu sprout extract showed the lowest isoflavone content of all the extracts, but schaftoside and ononin were only detected in sprouts; this is the first report of these compounds being present in kudzu, and they have been reported to have anti-obesity [33] and anti-inflammatory activities [34].

The extracts from the leaves, roots, sprouts, and stems of kudzu might serve as good sources for a variety of phytochemicals. Among the extracts, that of the leaves extract of kudzu would be a good source of genistin.

## 3. Materials and Methods

### 3.1. Materials

Fresh samples of the roots, sprouts, stems, and leaves of kudzu were collected from Daejeon, Korea, in 2016 by Professor Kyoung Yup Ji of Daeku Haany University. The botanical materials were identified by Professor Geung-joo Lee of Chungnam National University. Fresh samples were dried under the shade. The voucher specimen was deposited at the herbarium of Korea Institute of Oriental Medicine (KIOM201701018962, Daejeon, South Korea). Dried samples (100 g) were extracted with 1.5 L of 70% ethanol for 3 hours at reflux, and these extracts were concentrated under reduced pressure, freeze dried, and stored at 4 °C.

### 3.2. Chemicals

2,2-Amino-bis(3-ethylbenzothiazoline-6-sulfonic acid) diammonium salt, Folin-Ciocalteu reagent, 2,2-diphenyl-1-picrylhydrazyl (DPPH), tannic acid, epigallocatechin gallate (EGCG), puerarian, schaftoside, daidzin, genistin, ononin, daidzein, genistein, 3-[4,5-dimethylthiazol-2-yl]-2,5-diphenyl-tetralium bromide (MTT), Griess reagent, RIPA buffer, phosphatase inhibitor cocktail 3, protease inhibitor cocktail, aluminum chloride, 1.0 M nitrobenzene solution, and ammonium persulfate were purchased from Sigma-Aldrich (St. Louis, MO, USA). Ethanol, acetonitrile, and methanol were purchased from J. T. Baker (Deventer, Netherlands). Murine macrophage cells (RAW 264.7) were purchased from American Type Culture Collection (Manassas, VA, USA). Dulbecco’s modified Eagle medium (DMEM), and fetal bovine serum (FBS) were purchased from Lonza Co. Ltd. (Basel, Switzerland). Antibodies for actin, COX-2, and iNOS were purchased from Cell Signaling Technology (Danvers, MA, USA).

### 3.3. Total Phenolic Content Assay (TPC)

Total phenolic content was determined by a Folin–Ciocalteu spectrophotometric method [35]. Aliquots of each sample (0.05 mL) were added to 0.05 mL of freshly diluted Folin–Ciocalteu reagent. The mixtures were allowed to equilibrate for 3 min and were then mixed with 0.05 mL of aqueous Na_2_CO_3_ (0.7 M). After incubation at room temperature for 60 min, the absorbance of the solution was read at 730 nm. A standard curve was prepared from different concentrations (0.01–1.0 mg/mL) of tannic acid, and the results are expressed as mg tannic acid equivalents (TAE) per gram of extract.

### 3.4. DPPH Scavenging Activity Assay

The DPPH FRS activity was determined using a modified version of the method described by Liu [36]. The sample (0.01–1.0 mg/mL in water, 0.12 mL) was mixed with 0.06 mL of 0.45 mM ethanolic DPPH solution. The solution was shaken vigorously using a vortexer, and after subsequent incubation for 15 min at room temperature, the absorbance was measured at 517 nm. EGCG solutions at concentrations of 0.01–1.0 mg/mL were used as a positive control. The DPPH FRS activity was calculated using the following equation:DPPH FRS activity (%) = (1 − $\frac{\left(A3-A4\right)}{\left(A1-A2\right)}$) ×100A1: The value for solvent absorbance with DPPH solutionA2: The value for solvent absorbance without DPPH solutionA3: The value for sample absorbance with DPPH solutionA4: The value for sample absorbance without DPPH solution

### 3.5. ABTS Radical Scavenging Activity Assay

The ABTS radical scavenging activity was determined by the method described by Huang [37]. An aqueous solution of ABTS^+^ (7 mM) was oxidized with potassium peroxodisulfate (2.4 mM) for 24 h in the dark at 4 °C. The ABTS^+^ radical solution was diluted with ethanol, and the absorbance of the solution was measured at 734 nm. The samples (0.1 mL, 0.01–1.0 mg/mL in water) were mixed with 0.1 mL of dilute ABTS^+^ radical solution, the mixtures were shaken vigorously, equilibrated at 7 min at room temperature in the dark, and then the decrease in absorbance at 734 nm was measured. BHA (butylene hydroxy anisole) at 0.01–1.0 mg/mL was used as a positive control. The ABTS^+^ radical scavenging activity was calculated using the following equation:ABTS^+•^ FRS activity (%) = (1 − $\frac{\left(A3-A4\right)}{\left(A1-A2\right)}$) ×100A1: The value for solvent absorbance with ABTS^+^ radical solutionA2: The value for solvent absorbance without ABTS^+^ radical solutionA3: The value for sample absorbance with ABTS^+^ radical solutionA4: The value for sample absorbance without ABTS^+^ radical solution

### 3.6. Xanthine Oxidase Inhibitory Activity Assay

Xanthine oxidase (XO) activity was assayed by measuring uric acid formation using a spectrophotometer. The reaction was conducted with 50 mM sodium phosphate buffer (pH 7.6), 17.9 nM xanthine sodium salt, and 0.04 units of xanthine oxidase. The inhibition of the XO activity was measured by following the decrease in the uric acid formation at 295 nm. The test samples were dissolved in DMSO, and the reaction was initiated by the addition of 200 μg/mL xanthine and xanthine oxidase at 37 °C. The final concentration of DMSO (0.1%, *v*/*v*) did not interfere with the enzyme activity. The XO activity was evaluated by comparing the results of the test solutions with allopurinol as the positive control [38].

### 3.7. Cell Culture and Viability

The RAW 264.7 cell line was obtained from Korea Cell Line Bank (#No. 40071, Seoul, Korea) and cultured in DMEM supplemented with 10% FBS and 1% penicillin/streptomycin (100 U/mL) and incubated at 37 °C in 5% CO_2_. To determine cell viability, RAW 264.7 cells were seeded in a 96-well plate at a density of 3 × 10^4^ cells/well. After incubating overnight, the cells were treated with various concentrations of the sample compounds in the presence of 1 μg/mL LPS and incubated for an additional 24 h. Cell viability was determined by an MTT assay.

### 3.8. Immunoblotting Analysis

Murine RAW 264.7 macrophages (TIB-71) were obtained from American Type Culture Collection (Manassas, USA) and seeded in a 6-well plate at a concentration of 2.0 × 10^5^ cells/mL. After incubation overnight, the supernatant was removed and various concentrations of samples with 1 µg/mL LPS were added at 2 mL/well along with penicillin and FBS-free DMEM. After 24 h of incubation, the supernatants were collected and subjected to NO production analysis. The cells were washed twice with PBS, and the remaining liquid was completely removed. The cells were then lysed with 60 μL of RIPA buffer containing 0.1% protease inhibitor and centrifuged at 4 °C at 15,000 rpm for 15 min to get the supernatant. The supernatant was quantified using a Bradford assay with bovine serum albumin and heating at 95 °C for 10 min.

Cellular proteins (20 µg) were electrophoresed on 10% SDS-PAGE and then electro-transferred to a PVDF membrane. The membrane was blocked with 5% skim milk for 2 h and then blocked with the 1:1000 diluted primary antibody (specific monoclonal antibodies for iNOS, COX, and *β*-actin) for 2 h. The membranes were washed 3 times with Tris-buffered saline (TBS) with 0.1% Tween 20 for 10 min. The membranes were then blocked with a 1:1000 diluted secondary antibody (anti-rabbit IgG for iNOS and COX-2 or anti-mouse IgG for *β*-actin) at room temperature for 1 hour. After that, the membranes were washed 3 times with TBS with 0.1% Tween 20 for 10 min. Protein bands were detected by a Western imaging system (CAS-400SM, Davinch-K Co. Ltd., Korea). The iNOS and COX immunoblot signals were compared with those of *β*-actin, and the relative protein expression was calculated.

### 3.9. NO Production Assay

As detailed above, the nitrite concentration in the cell supernatant was used as an indicator of NO production using the Griess reagent (Sigma). Briefly, 100 μL of each supernatant was mixed with 100 μL of Griess reagent (1% sulfanilamide in 5% phosphoric with 0.1% naphthylethylenediamine dihydrochloride in water) in a 96-well plate. The mixtures were incubated at room temperature for 10 min, and then the absorbance of each well was determined at 540 nm using a microplate reader. The amount of nitrite in each sample was back-calculated from a sodium nitrite calibration curve (0–100 μM).

### 3.10. High-Performance Liquid Chromatography Analysis

A Waters Acquity UPLC system equipped with a quaternary pump, auto-sampler, and photodiode array detector with Acquity UPLC BEH C18, 100 × 2.1 mm, 1.7 μm was used for UPLC analysis (Waters, MA, USA). The mobile phase consisted of water containing 0.1% formic acid (solvent A) and acetonitrile (solvent B) with gradient elution (a linear gradient from 5% to 15% B in 10 min, a linear gradient from 15% to 20% B in 5 min, a linear gradient from 20% to 30% B in 5 min, a linear gradient from 30% to 100% B in 2 min, a linear gradient from 100% to 5% B in 1 min, and finally, isocratic elution for 2 min). The flow rate was 0.5 mL/min, and quantitative measurements were made at 245 nm. Detection was by a mass spectrometry (MS) equipped with an electrospray ionization (ESI) source in positive modes. The instrument parameters were a capillary voltage of 0.8 kV and cone voltage of 15 V. The probe temperature was 600 °C. MS data was collected in full-scan mode ranging from 220 to 600. The desolvation gas was Nitrogen (600 L/h).

### 3.11. Statistical Analysis

All data are expressed as the mean ± standard deviation (SD). The results were obtained from at least three independent experiments. Statistical analyses were conducted using one-way analysis of variance (ANOVA) followed by Tukey’s test for multiple comparisons (GraphPad Prism 7.0, GraphPad Software Inc., San Diego, USA). *p* values less than or equal to 0.05 were considered significant.

## 4. Conclusions

Among the extracts from the roots, leaves, stems, and sprouts of kudzu, the leaf extract showed the highest antioxidant activities in the DPPH, ABTS, and xanthine oxidase inhibition assays, and it showed the highest anti-inflammatory activity based on suppression of NO production in RAW 264.7 macrophage cells stimulated with LPS. Its anti-inflammatory activity was confirmed by its ability to suppress iNOS and COX-2 expression. The leaf extract was found to have remarkably high contents of genistin and daidzin compared to those of the other aerial parts and root extracts. This study suggests that the leaves of kudzu are a good source of isoflavones and may be useful as a functional or medicinal use related to inflammation, bone loss, metabolic disorders, and aging arising from oxidative stress.

## Figures and Tables

**Figure 1 molecules-24-00912-f001:**
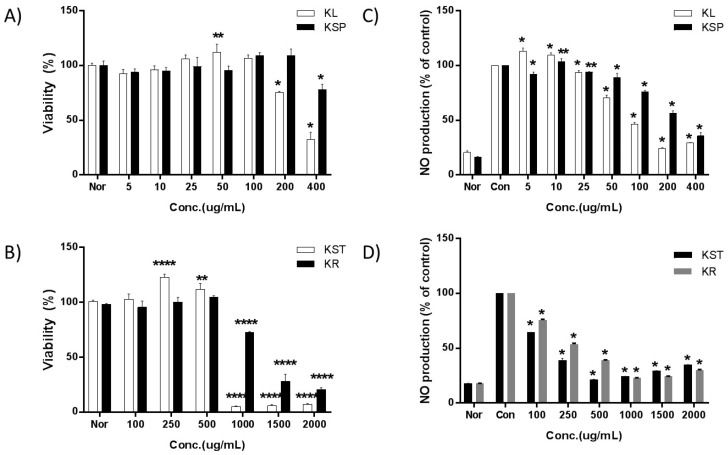
Effects by extraction parts of kudzu on the viability and NO levels of RAW 264.7 macrophages. A density of 1 × 10^4^ cells/well of macrophages were seeded in a 96-well plate and incubated with various concentrations of each extract for 24 h and cell viability was determined by MTT assay. A density of 1 × 10^4^ cells/well of macrophages were seeded in a 96-well plate and the various concentrations of sample and LPS (1 μg/mL) were treated with DMEM (without penicillin and fetal bovine serum (FBS)) to 0.2 mL/well and incubated for 24 h. Untreated samples without LPS treatment were negative controls. Each value is expressed as mean ± SD of three independent experiments. *; *p* < 0.01, **; *p* < 0.005, ***; *p* < 0.001 and ****; *p* < 0.0005 versus culture media without sample which acts as a control.

**Figure 2 molecules-24-00912-f002:**
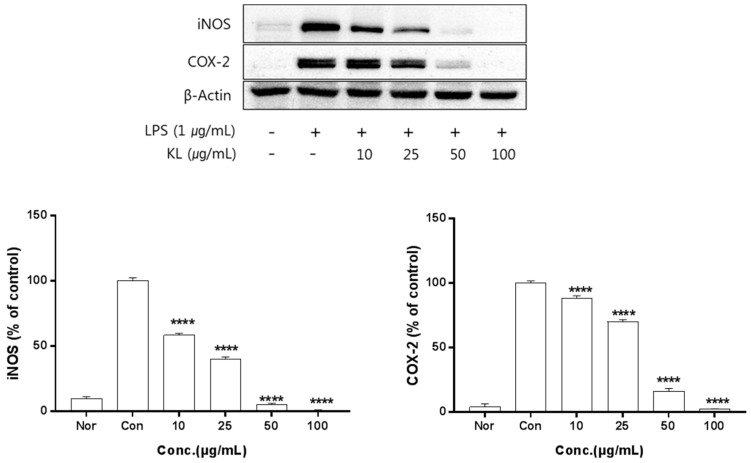
Inhibitory effects of KL on iNOS and COX-2 production in LPS-stimulated Raw 264.7 cells. Equal amounts of cell lysates (30 μg) were subjected to electrophoresis and analysed for inducible nitrix oxide synthase (iNOS) and COX-2 production by Western blot. Raw 264.7 cells were incubated with LPS (1 μg/mL) and KL extract for 24 h. Each value is expressed as mean ± SD of three independent experiments. **** *p* < 0.0005 as analyzed by Duncan’s multiple range test. Nor: normal; Con: control.

**Figure 3 molecules-24-00912-f003:**
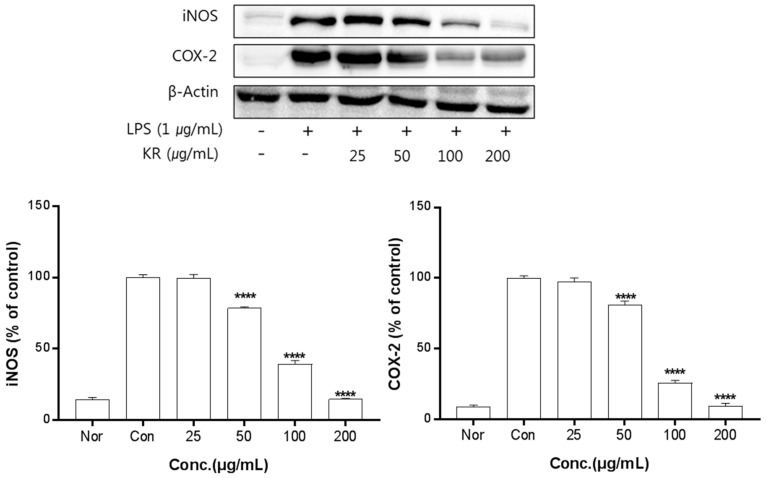
Inhibitory effects of KR on iNOS and COX-2 production in LPS-stimulated Raw 264.7 cells. Equal amounts of cell lysates (30 μg) were subjected to electrophoresis and analysed for iNOS and COX-2 production by Western blot. Raw 264.7 cells were incubated with LPS (1 μg/mL) and KR extract for 24 h. Each value is expressed as mean ± SD of three independent experiments. **** *p* < 0.0005 as analyzed by Duncan’s multiple range test.

**Figure 4 molecules-24-00912-f004:**
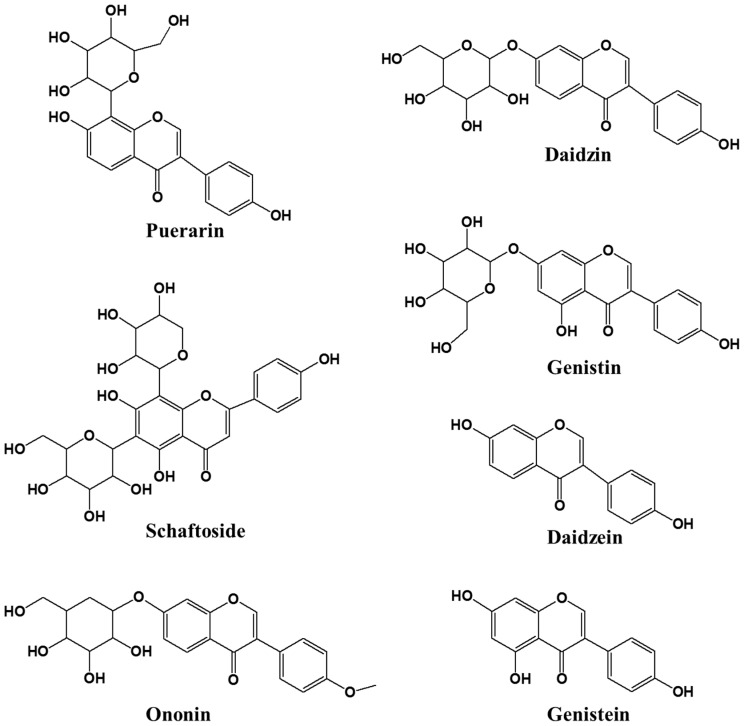
Chemical structures of seven isoflavonoids identified from each part of kudzu.

**Figure 5 molecules-24-00912-f005:**
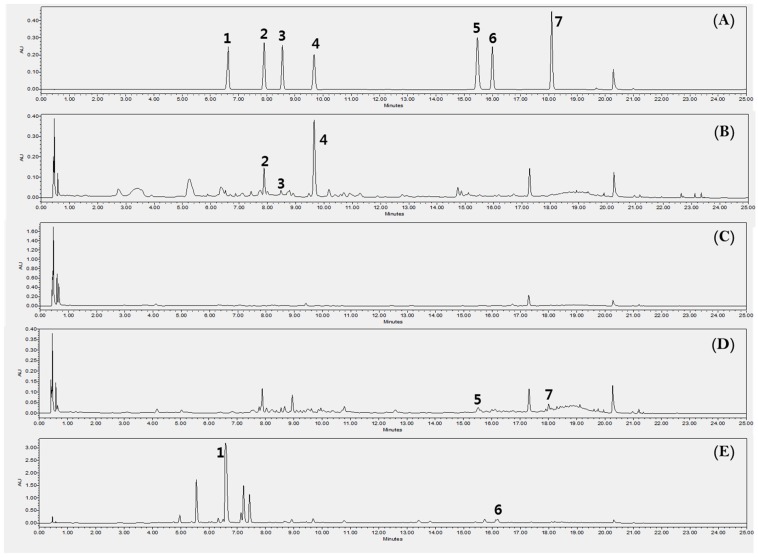
Ultra Performance Liquid Chromatography with Diode-Array Detection (UPLC-DAD) chromatogram for the quantification of the compounds 1–7 in each extract of the part of kudzu at 245 nm. A BEH column (Waters, USA), C18 (2.1 × 100 mm i.d.; 1.7 µm) was used. The gradient program with 0.1% formic acid with water (A) and acetonitrile (B) flowing at a rate of 0.5 mL/min was: 5–15% B (0–10min), 15–20% B (10–15 min), 20–30% B (15–20 min), 30–100% B (20–22 min), 100–5% B (22–23 min), 5–5% B (23–25 min). The volume of injection was 5 µL. 1: Puerarin (6.6 min), 2: Daidzin (7.9 min), 3: Schaftoside (8.6 min), 4: Genistin (9.8 min), 5: Ononin (15.5 min), 6: Daidzein (16.0 min), 7: Genistein (18.1 min). (**A**) Reference standards, (**B**) KL, (**C**) KSP, (**D**) KST, (**E**) KR.

**Table 1 molecules-24-00912-t001:** Half maximal Free radical scavenging (FRS_50_) value by extraction part of kudzu for antioxidant systems.

Sample	FRS_50_ (μg/mL)	
	DPPH FRS Activity	ABTS^+^• FRS Activity
KL	437 ± 10.9 ^b^	121 ± 6.6 ^b^
KST	1,136 ± 14.2 ^e^	455.3 ± 17.1 ^e^
KR	582 ± 16.4 ^c^	138.0 ± 2.7 ^c^
KSP	755 ± 8.6 ^d^	341.7 ± 1.4 ^d^
Control	17 ± 0.3 ^a^	5.5 ± 1.0 ^a^

Each value in the table is represented as mean ± SD (*n* = 3). Means not sharing the same letter are significantly different (LSD) at *p* < 0.05 probability level in each column. KL: Kudzu leaf extract; KST: Kudzu stem extract; KR: Kudzu root extract; KSP: Kudzu sprout extract.

**Table 2 molecules-24-00912-t002:** The content of seven major compounds by extraction part of kudzu.

	Content (mg/g)							
	1	2	3	4	5	6	7	SUM
KL		5.5 ± 0.3	0.8 ± 0.0	60.8 ± 1.1				66.3 ± 1.42
KST	0.6 ± 0.1	0.8 ± 0.1		0.1 ± 0.0				2.3 ± 0.2
KR	67.1 ± 2.2			4.3 ± 0.3	2.0 ± 0.2			73.4 ± 2.7
KSP		*		*	*	*	*	*

The quantification of the compounds 1-6 in each extracts of the part of kudzu at 245 nm. 1: Puerarin (6.6 min), 2: Daidzin (7.9 min), 3: Schaftoside (8.6 min), 4: Genistin (9.8 min), 5: Ononin (15.5 min), 6: Daidzein (16.0 min), 7: Genistein (18.1 min), and * means that compound was detected below the minimum quantification limit. Each value in the table is represented as mean ± SD (*n* = 3).

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
