# Peer review of "Comparison among Activities and Isoflavonoids from Pueraria thunbergiana Aerial Parts and Root"

_molecules, 2019, doi:10.3390/molecules24050912_

Round 1

Reviewer 1 Report

The paper reports the results of a study on the chemical components of the aerial parts of Pueraria thunbergiana, and the antioxidant properties of the extracts. The paper is well written with sufficient details in the experimental part for reproductive studies. The major concern is identification of the 7 major constituents: retention time and UV spectra are not the standard for identification purposes: it would have been required to compare also the MS-MS spectra of the compounds with those of reference products. There is also the lack of a general scheme summarizing all identified compounds structures: it should be added in a revised version of the manuscript. A few errors must be corrected as suggested below.

- in the abstract, 1st line write : medicines for many...

- page 2 lines 65 and 70 write : root extract (and roots extracts)

- page 6, add a scheme with the structure of the 7 identified compounds

- page 8, define the Folin-Ciocalteu reagent

In conclusion, the paper could be accepted for publication after corrections as suggested above.

Author Response

Point 1: The paper reports the results of a study on the chemical components of the aerial parts of Pueraria thunbergiana, and the antioxidant properties of the extracts. The paper is well written with sufficient details in the experimental part for reproductive studies. The major concern is identification of the 7 major constituents: retention time and UV spectra are not the standard for identification purposes: it would have been required to compare also the MS-MS spectra of the compounds with those of reference products. There is also the lack of a general scheme summarizing all identified compounds structures: it should be added in a revised version of the manuscript. A few errors must be corrected as suggested below.

 Response 1: The structures of 7 compounds were added to Fig. 4, the LC-MS analysis was performed, and the results were presented at line 172 page 6 and line 310 page10 as below.

[Result] Each 7 components were identified by confirmation on the positive ionized mass spectrum as follows; Puerarin ([M+H]+ 417.29), Daidzin ([M+H]+ 417.32), Schaftoside ([M+H]+ 565.41), Genistin ([M+H]+ 433.29), Ononin ([M+H]+ 431.31), Daidzein ([M+H]+ 255.17), Genistein ([M+H]+ 271.14).

[Method] Detection was used a mass spectrometry (MS) equipped with an electrospray ionization (ESI) source in positive modes. The instrument parameters were a capillary voltage of 0.8 kV and cone voltage of 15 V. The probe temperature was 600 °C. MS data was collected in full-scan mode ranging from 220 to 600. The desolvation gas was Nitrogen (600 L/h).

Point 2: in the abstract, 1st line write : medicines for many...

Response 2: It has been done at line 15 page 1.

Point 3: page 2 lines 65 and 70 write : root extract (and roots extracts)

Response 3: It has been done.

Point 4: add a scheme with the structure of the 7 identified compounds

Response 4: The structures of 7 compounds were added to Fig. 4.

Point 5: define the Folin-Ciocalteu reagent

Response 5: Folin-Ciocalteu is an assay regularly used to predict total phenolics. This method was developed by Folin and Ciocalteu in 1927. This method uses molybdotungstophosphoric heteropolyanion reducting reagent which indirectly detects phenolics and is reaction using the color change of molybdenum by the phenolics.

Reviewer 2 Report

The manuscript describes a comparison among the constituents and their antioxidant and anti-inflammatory activities of the aerial parts of Kudzu and those reported for the root of the same species. These results will help to add value to this species.

Please consider modifying the title, for example: "Comparison among activities and isoflavonoids from Pueraria thunbergiana aerial parts and root"

In general, the manuscript is well written, with some minor corrections highlighted in yellow in the enclosed pdf file. You should review all the significant figures and change the IC for FRS term for the DPPH and ABTS tests.

You must review the list of references, using the MDPI style. Please see "https://www.mdpi.com/authors/references".

With these considerations, the article is recommended to be published in Molecules.

Author Response

Point 1: Please consider modifying the title, for example: "Comparison among activities and isoflavonoids from Pueraria thunbergiana aerial parts and root"

Response 1: It has been done.

Point 2: In general, the manuscript is well written, with some minor corrections highlighted in yellow in the enclosed pdf file. You should review all the significant figures and change the IC for FRS term for the DPPH and ABTS tests.

Response 2: It has been done according to your comments.
